# Diagnostic performance of abbreviated non-contrast liver MRI for detecting synchronous colorectal liver metastases

Puwitch Charoenchue👁*, Kamon Rajchakom, Amonlaya Amantakul, Suwalee Pojchamarnwiputh, Chanon Thanaphakpaisarn, Wittanee Na Chiangmai, Tanin Lertsiriladakul

Department of Radiology, Faculty of Medicine, Chiang Mai University, Chiang Mai, Thailand

* puwitch.c@cmu.ac.th

## Abstract

### Background

Colorectal cancer (CRC) is a leading malignancy, and colorectal liver metastases (CRLM) are common. Accurate detection is critical for staging and treatment planning. Although contrast-enhanced MRI is the reference standard, its use is limited by cost, longer examination time, and contraindications such as renal impairment or prior severe allergic reactions. This study aimed to evaluate the diagnostic performance of an abbreviated non-contrast liver MRI protocol for detecting CRLM and to compare performance between experienced and in-training readers.

### Materials and methods

In this retrospective study, 87 patients with CRC who underwent liver MRI between January 2014 and March 2024 were included. The abbreviated non-contrast protocol comprised T1-weighted imaging (T1WI), T2-weighted imaging (T2WI), and diffusion-weighted imaging (DWI). Two independent readers evaluated the images using a 4-point confidence scale, which was dichotomized for analysis. Sensitivity, specificity, predictive values, and accuracy were calculated, and interobserver agreement was assessed using weighted kappa. Contrast-enhanced MRI served as the reference standard.

### Results

The abbreviated non-contrast MRI demonstrated high diagnostic performance. Sensitivity and specificity were 94.1% and 88.7% for the abdominal radiologist, and 91.2% and 96.2% for the oncologic imaging fellow. The area under the receiver operating characteristic curve (AUROC) was 0.93 and 0.94, respectively, with no significant

which permits unrestricted use, distribution, and reproduction in any medium, provided the original author and source are credited.

**Data availability statement:** All relevant data are within the manuscript and its Supporting Information files. The de-identified dataset used for statistical analysis is provided as S1 Data (.dta and.xlsx format). Due to privacy and ethical restrictions, raw imaging data cannot be publicly shared; however, all data necessary to reproduce the results are included.

**Funding:** The author(s) received no specific funding for this work.

**Competing interests:** The authors have declared that no competing interests exist.

difference between readers ($p = 0.494$). Interobserver agreement was substantial (weighted $\kappa = 0.74$).

## Conclusion

Non-contrast MRI demonstrated reliable diagnostic performance with substantial interobserver agreement for CRLM detection, suggesting its potential clinical applicability. While contrast-enhanced MRI remains the standard, non-contrast MRI may be a feasible alternative in select cases. Further large-scale validation is necessary before broader clinical adoption.

## 1. Introduction

Colorectal cancer (CRC) ranks among the most common cancers globally [1], with a substantial number of patients either presenting with colorectal liver metastases (CRLM) at initial diagnosis or developing them during the progression of the disease [2–4]. The mortality rate from CRC remains alarmingly high and is continuing to rise, particularly in developing countries [5,6]. The precise identification of liver metastases during the initial staging is essential for the formulation of an effective treatment plan, as it impacts surgical decisions, options for local or systemic therapy, and the overall prognosis [7,8].

Contrast-enhanced magnetic resonance imaging (MRI) is regarded as the gold standard for hepatic imaging, owing to its exceptional soft-tissue contrast and lesion-characterization capabilities [9–12]. Nonetheless, using contrast agents entails higher costs, longer scan times, and potential contraindications for individuals with kidney disease or gadolinium-based contrast agent (GBCA) allergies. Additionally, concerns regarding gadolinium retention in tissues, the risk of nephrogenic systemic fibrosis (NSF) in patients with renal impairment, and potential allergic reactions further limit its widespread use [13]. These challenges underscore the need for alternative imaging approaches, such as non-contrast MRI, which may offer a safer and more accessible option for detecting colorectal liver metastasis. Recent advances in MRI techniques, particularly diffusion-weighted imaging (DWI) and T2-weighted imaging (T2WI), have improved the detection of liver tumors, such as hepatocellular carcinoma and especially CRLM [14–17]. Several studies have demonstrated that non-contrast MRI, particularly when incorporating DWI, provides high sensitivity and specificity in detecting liver metastases, potentially serving as an alternative to contrast-enhanced MRI in specific clinical scenarios [16,18,19].

However, most prior studies have been conducted in lesion-based analyses or selected patient populations, often focusing on protocol comparisons rather than clinically relevant decision-making. In routine practice, imaging is typically used to determine the presence or absence of metastasis at the patient level, which has direct implications for staging and treatment planning. Therefore, the role of abbreviated non-contrast MRI in real-world staging settings remains incompletely defined.

The primary objective of this study was to evaluate the diagnostic performance of an abbreviated non-contrast liver MRI protocol for detecting CRLM in patients undergoing initial staging of CRC. The secondary objective was to compare diagnostic performance between an experienced abdominal radiologist and an oncologic imaging fellow. We hypothesized that abbreviated non-contrast MRI would demonstrate high diagnostic performance and that diagnostic performance would be comparable between experienced and in-training readers, supporting its use in selected clinical scenarios.

## 2. Materials and methods

### 2.1. Study design and population

This retrospective study was certified as exempt from full ethics review by the Research Ethics Committee of the Faculty of Medicine, Chiang Mai University (Exemption No. 0584/2024; Study code RAD-2567–0584). The requirement for informed consent was waived. Data were accessed for research purposes between November 01, 2024, and June 30, 2025, and only anonymized datasets were available to the investigators. Data were collected from January 01, 2014, to March 31, 2024, from an imaging database using search terms "colorectal" or "rectal" combined with "cancer," "carcinoma," or "CA" in MRI reports.

The sample size was determined using Cochran's method [20], with an assumed population proportion of 0.20, a margin of error of 5%, and a 95% confidence interval. Substituting these values, the required sample size was 62 patients, with an adjustment for a 10% dropout rate, leading to a final estimated requirement of 69 patients. Although formal sample size determination for diagnostic accuracy studies is challenging, the final sample size exceeded the minimum estimated requirement. It was comparable to that of prior studies evaluating abbreviated liver MRI protocols for colorectal liver metastases [18,21,22].

A total of 454 patients with colorectal cancer (CRC) and suspected colorectal liver metastases (CRLM) at primary staging were identified, of whom 104 underwent abdominal MRI, including liver imaging, as part of the staging process. All patients had pathologically confirmed CRC via biopsy or surgery.

The abbreviated non-contrast liver MRI was evaluated as the test under investigation, with full contrast-enhanced MRI serving as the reference standard. Inclusion criteria required patients to be (i) 18 years or older, (ii) diagnosed with CRC confirmed by histopathology, and (iii) undergoing a full liver MRI protocol at primary staging. Seventeen patients were excluded, including 16 with other active malignancies and one with a prior liver ablation, resulting in a final study population of 87 patients. The patient selection process is illustrated in S1 Fig.

### 2.2. Reference test

All MRIs were acquired using the following MRI systems:

3.0-Tesla MRI systems:

1. MAGNETOM Skyra (Siemens Healthineers, Erlangen, Germany) – 18-channel phased-array coil

2. SIGNA Pioneer (GE Healthcare, Chicago, IL, USA) – 30-channel phased-array coil

1.5-Tesla MRI systems:

3. MAGNETOM Altea (Siemens Healthineers, Erlangen, Germany) – 12-channel phased-array coil

4. SIGNA HDxt with XP (GE Healthcare, Chicago, IL, USA) – 12-channel phased-array coil

The full liver MRI protocol included the following sequences:

1. Axial T2-weighted images (T2WI) with spectral fat suppression, acquired using fast spin echo-based techniques, including fast recovery fast spin echo (FRFSE) and heavily T2-weighted sequences.

2. Coronal T2WI using single-shot techniques, including half-Fourier acquisition single-shot turbo spin echo (HASTE; Siemens Healthineers) and single-shot fast spin echo (SSFSE; GE Healthcare).

3. Axial 3D T1-weighted images (T1WI) were acquired using frequency-selective fat suppression based on spectral presatration with inversion recovery (SPIR) acquired using vendor-specific techniques, including liver acquisition with volume acceleration (LAVA; GE Healthcare) and volumetric interpolated breath-hold examination (VIBE; Siemens Healthineers).

4. Diffusion-weighted imaging (DWI) with b-values of 100, 500, and 800 s/mm$^2$

For contrast-enhanced imaging, gadolinium-based contrast agents were administered intravenously at 2–2.5 mL/s, followed by a 20-mL saline flush. The contrast agents used were gadoxetate disodium (Primovist, Bayer Healthcare, Leverkusen, Germany) at a dose of 0.025 mmol/kg and gadobenate dimeglumine (MultiHance, Bracco Imaging, Milan, Italy) at a dose of 0.1 mmol/kg.

For dynamic contrast-enhanced imaging, fat-suppressed T1WI were acquired across multiple phases, including the arterial phase (20–25 seconds), the portal venous phase (60 seconds), and a 3-minute delayed phase. The hepatobiliary phase (HBP) was captured 20 minutes after the injection of Primovist and 60 minutes after the injection of MultiHance.

Two abdominal radiologists with more than ten years of experience reviewed and validated all reference MRI studies by consensus.

## 2.3. Reading session

The provided test images for the readers included coronal fast T2WI, axial fat-suppressed T2WI, axial fat-suppressed T1WI, and axial DWI with b-values of 0, 500, and 800 s/mm², as illustrated in Fig 1. The abbreviated liver MRI datasets were extracted from the reference MRI protocol and independently reviewed by an abdominal radiologist with eight years of experience (Reader 1) and an oncologic imaging fellow (Reader 2). They were blinded to the radiologic results, histopathologic data, and other sequences in the complete MRI protocol that were not included in the abbreviated protocol.

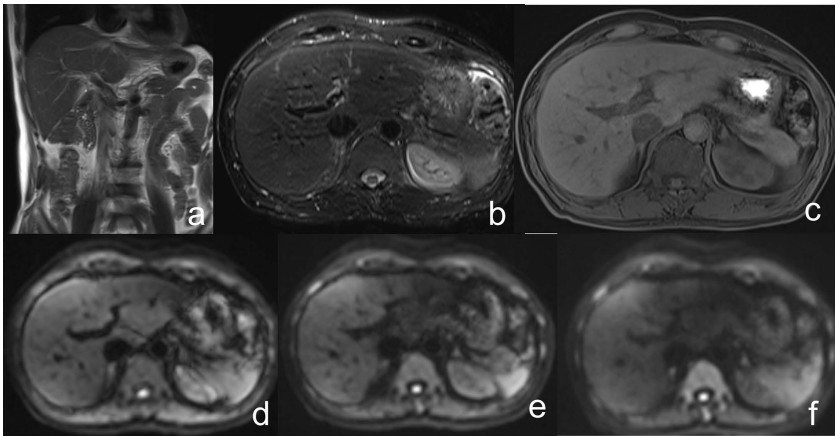

**Fig 1. Abbreviated non-contrast liver MRI protocol used for reader evaluation.** (a) coronal T2WI, (b) axial fat-suppressed T2WI, (c) axial fat-suppressed 3D T1WI, and (d–f) axial DWI with b-values of 100, 500, and 800 s/mm².

Image interpretation included the number of liver lesion(s) and a 4-point confidence score for the presence of metastasis, defined as follows: 0 = definitely not metastasis; 1 = unlikely metastasis; 2 = likely metastasis; and 3 = definitely metastasis.

For diagnostic performance analysis, results were dichotomized into "metastasis" (confidence scores of 2 or 3) and "non-metastasis" (confidence scores of 0 or 1).

### 2.4. Statistical analysis

Continuous variables were summarized as mean ± standard deviation (SD), and categorical variables were presented as frequencies and percentages.

To assess the diagnostic performance of the abbreviated non-contrast liver MRI, sensitivity, specificity, positive predictive value (PPV), negative predictive value (NPV), and overall accuracy were calculated for each reader using contrast-enhanced MRI as the reference standard. For these analyses, diagnostic confidence scores were dichotomized into positive (scores 2–3) and negative (scores 0–1), reflecting clinically relevant decision thresholds.

Receiver operating characteristic (ROC) analysis was performed using the 4-point diagnostic confidence scores (0–3) as an ordinal test variable to evaluate the ability of the abbreviated MRI to discriminate between patients with and without CRLM. The area under the ROC curve (AUROC) was calculated for each reader as a measure of overall discriminatory performance. Differences in AUROC between readers were assessed using paired ROC curve comparison.

Inter-reader agreement between the two radiologists was evaluated using Cohen's kappa ($\kappa$) statistic with quadratic weights, to account for the ordinal nature of the diagnostic confidence scores. $\kappa$ values were interpreted as follows: < 0.20 = poor agreement; 0.21–0.40 = fair agreement; 0.41–0.60 = moderate agreement; 0.61–0.80 = substantial agreement; and > 0.81 = almost perfect agreement. The McNemar test was used to compare paired proportions, including sensitivity and specificity, between two readers.

All statistical analyses were performed using STATA version 19.5 (StataCorp LLC, College Station, TX, USA). A $p$-value of < 0.05 was considered statistically significant. The original dataset used for analysis is provided as Supporting Information (S1 Data).

## 3. Results

### 3.1. Patient characteristics

A total of 87 patients were included in the analysis, comprising 51 men (58.6%) and 36 women (41.4%). The mean age of the study population was 69 years (±12.9), with a range of 38–102 years. Regarding tumor staging, most patients were classified as stage 3 (26.4%) or stage 4 (47.1%), while a smaller proportion were in stages 0–2, as shown in Fig 2. CRLM was detected in 39.1% of patients (34 out of 87). Among stage 4 patients, those without liver metastases had metastases to other sites, including non-regional lymph nodes, bone, lung, or peritoneal carcinomatosis. The comparison of baseline characteristics between patients with and without CRLM is summarized in Table 1.

### 3.2. Diagnostic performance of abbreviated non-contrast MRI

The diagnostic performance of each observer in detecting CRLM using abbreviated non-contrast liver MRI is summarized in Table 2. Reader 1 demonstrated a sensitivity of 94.1% and a specificity of 88.7%, whereas Reader 2 achieved a sensitivity of 91.2% and a specificity of 96.2%. The PPV was 84.2% for Reader 1 and 93.9% for Reader 2, while the NPV was 95.9% and 94.4%, respectively. Overall accuracy was 90.8% for Reader 1 and 94.3% for Reader 2, with no statistically significant difference between readers ($p = 0.387$).

The discriminatory performance of the abbreviated MRI protocol was excellent for both readers, with an AUROC of 0.93 for Reader 1 and 0.94 for Reader 2. There was no statistically significant difference between the two ($p = 0.494$), as shown in S2 Fig. Detailed contingency tables for each reader are provided in S1 Table.

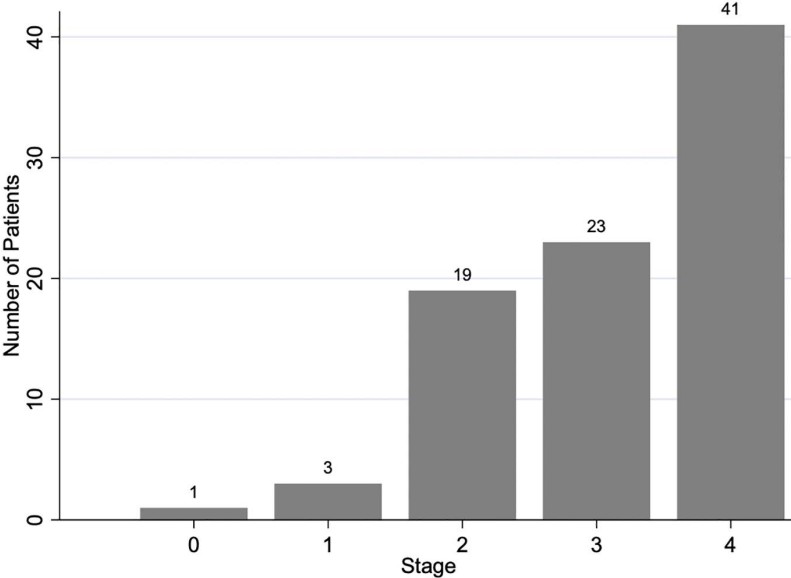

**Fig 2. Distribution of colorectal cancer stages in the study population.**

**Table 1. Patient characteristics stratified by liver metastasis (CRLM) status.**

| Characteristics | Total (*n*=87) | CRLM present (*n*=34) | No CRLM (*n*=53) |
|---|---|---|---|
| Age (years) | 69.0±12.9 | 68.9±9.9 | 69±14.6 |
| Sex, *n* (%) | | | |
| Male | 51 (58.6%) | 21 (61.8%) | 30 (56.6%) |
| Female | 36 (41.4%) | 13 (38.2%) | 23 (43.4%) |
| Cirrhosis | 6 (6.9%) | 0 (0.0%) | 6 (11.3%) |
| Hypertension | 29 (33.3%) | 11 (32.4%) | 18 (34.0%) |
| Dyslipidemia | 18 (20.7%) | 8 (23.5%) | 10 (18.9%) |
| Diabetes mellitus | 11 (12.6%) | 5 (14.7%) | 6 (11.3%) |
| Chronic hepatitis B | 2 (2.3%) | 0 (0.0%) | 2 (3.8%) |
| Other underlying diseases | 22 (25.3%) | 8 (23.5%) | 14 (26.4%) |
| Total protein, g/dL | 7.3±0.7 | 7.5±0.5 | 7.2±0.8 |
| Albumin, g/dL | 4.0±0.7 | 4.0±0.5 | 4.0±0.5 |
| Total bilirubin, mg/dL | 0.5±0.7 | 0.4±0.2 | 0.6±0.8 |
| Direct bilirubin, mg/dL | 0.3±0.4 | 0.2±0.1 | 0.3±0.5 |
| AST, U/L | 24.3±12.1 | 24.2±13.6 | 24.3±11.4 |
| ALT, U/L | 19.1±11.7 | 18.6±8.8 | 19.4±13.4 |
| ALP, U/L | 92±57 | 105.7±83.5 | 83.3±28.6 |

Data are presented as mean±standard deviation or number (percentage).

### 3.3. Interobserver agreement

Interobserver agreement between the abdominal radiologist and the oncologic imaging fellow was evaluated using Cohen's kappa statistics. Because of the ordinal nature of the 4-point confidence scale, quadratic weighted kappa was

**Table 2. Comparison of diagnostic performance between readers.**

| Metric | Reader 1 | Reader 2 | p-value |
|---|---|---|---|
| Sensitivity (%) | 94.1 (80.3–99.3) | 91.2 (76.3–98.1) | 0.457 |
| Specificity (%) | 88.7 (77.0–95.7) | 96.2 (87.0–99.5) | 0.059 |
| Positive Predictive Value (PPV) (%) | 84.2 (68.7–94.0) | 93.9 (79.8–99.3) | 0.040 |
| Negative Predictive Value (NPV) (%) | 95.9 (86.0–99.5) | 94.4 (84.6–98.8) | 0.647 |
| Accuracy (%) | 90.8 | 94.3 | 0.387 |
| AUROC | 0.93 (0.87–0.98) | 0.94 (0.89–1.00) | 0.494 |

primarily used. The weighted kappa was 0.74, indicating substantial agreement ($p < 0.001$), while the unweighted kappa was 0.88, reflecting almost perfect agreement. The overall percent agreement was 94.3%.

## 4. Discussion

Our study evaluated the diagnostic performance of abbreviated non-contrast liver MRI in detecting CRLM at the primary staging of CRC. The results demonstrated that both readers exhibited high sensitivity, specificity, and accuracy in detecting CRLM, with strong inter-reader agreement. Although the abdominal radiologist had slightly higher sensitivity, the difference was not statistically significant. The AUROC was high for both readers, with no statistically significant difference between them, indicating that abbreviated non-contrast MRI may be a reliable alternative for CRLM detection. Although histopathology is the gold standard, contrast-enhanced MRI interpreted by experienced radiologists is widely accepted for CRLM detection and treatment planning, particularly when surgical confirmation is unavailable.

The distribution of tumor staging in our cohort showed that the majority of patients were in stage 3 or stage 4, which aligns with previous studies reporting that a significant proportion of CRC patients present with advanced disease at initial diagnosis [10]. The prevalence of liver metastasis at the primary staging was approximately 15–25% [23]. Our investigation demonstrated a prevalence of 39.1%, which might be attributed to the inclusion of selective MRI studies. In our institution, computed tomography (CT) scans served as the primary modality for metastatic staging, incorporating both contrasted chest and abdominopelvic CT. In the absence of clinical or imaging indicators suggestive of metastasis, MRI was not performed. Only cases exhibiting suspicious metastatic features on CT or presenting with significantly elevated serum carcinoembryonic antigen (CEA) levels were referred for MRI evaluation. This approach is generalized in most centers [9]. Consequently, our study reported a higher prevalence of metastasis compared to other cross-sectional studies [23,24]. Given the higher prevalence of CRLM in our cohort, caution is warranted when interpreting prevalence-dependent measures such as PPV and NPV. Notably, in real-world settings with lower prevalence, the proportion of true negative cases would increase, and overall accuracy may be expected to be even higher, assuming similar specificity.

Most studies evaluating abbreviated liver MRI for detecting CRLM have used gadolinium-based contrast agents, including hepatobiliary contrast, and have shown improved diagnostic performance compared with non-contrast MRI [21,22,25,26]. Our study aims to assess non-contrast methods to determine whether T2WI and DWI are adequate for diagnosing CRLM in most cases. This may help reduce the use of GBCAs. The recent study further exemplifies the strong performance and cost-effectiveness of a non-contrast approach [18].

When comparing our findings with previous studies evaluating non-contrast MRI for diagnosing CRLM, our results are consistent with reports suggesting that non-contrast MRI has high diagnostic performance, particularly when DWI is included in the protocol [16,18,19]. Some studies have demonstrated comparable accuracy between non-contrast and contrast-enhanced MRI, reinforcing the potential utility of a non-contrast approach in specific clinical scenarios [19,21]. However, most prior studies were conducted in lesion-based or protocol-comparison settings, often involving selected populations with known or suspected lesions. In contrast, our study was designed using a patient-based approach in a

real-world staging context, reflecting the clinically relevant task of determining the presence or absence of CRLM at the patient level. Therefore, our findings extend the existing literature by demonstrating that an abbreviated non-contrast MRI protocol can achieve high diagnostic performance within a practical clinical workflow.

The abbreviated non-contrast liver MRI protocol selects T2WI and DWI as the primary sequences based on their high sensitivity. T2WI provides high soft-tissue contrast, allowing the delineation of liver lesions against the surrounding parenchyma. Liver metastases frequently demonstrate moderate to high signal intensity on fat-suppressed T2WI, which enhances lesion conspicuity even without contrast administration [27]. This characteristic makes T2WI a crucial sequence for detecting metastatic deposits. DWI enhances lesion detection by exploiting the restricted diffusion properties of malignant tumors. CRLM typically exhibits high signal intensity on high b-value DWI due to their high cellularity and reduced extracellular space, with corresponding low signal intensity on ADC maps, thereby differentiating them from benign lesions such as cysts [28,29]. Several studies have demonstrated that DWI can improve sensitivity for CRLM detection [30].

False-positive (FP) and false-negative (FN) findings were further analyzed to better understand the limitations of the abbreviated non-contrast MRI protocol (S2 Table). Most FP cases were related to benign lesions, particularly hepatic cysts, which demonstrated mild diffusion restriction on high-b-value DWI and may mimic metastatic lesions. Additional FP cases included other primary hepatic malignancies, such as hepatocellular carcinoma (HCC). Representative examples are shown in  Figs 3–5. In contrast, FN cases were predominantly small metastases that were either below the spatial resolution of the abbreviated protocol or exhibited only subtle diffusion restriction, as illustrated in Fig 6. These findings highlight a key limitation of DWI, where both benign lesions and small metastases may present with overlapping diffusion characteristics.

Although mucinous colorectal cancers may present diagnostic challenges due to overlapping imaging features with cystic lesions, no specific subgroup analysis was performed in this study, and this remains an area for future investigation.

Despite the good to excellent diagnostic performance of non-contrast MRI in this study, several limitations should be acknowledged. First, the sample size was relatively limited, although sufficient to achieve statistical power. Second, this was a retrospective study, and the study population was selected based on clinical suspicion or prior CT findings, which may introduce selection bias and result in a higher prevalence of CRLM than in the general population. This may affect the interpretation of prevalence-dependent measures such as PPV and NPV, although sensitivity, specificity, AUROC, and interobserver agreement remain valid.

Third, the abbreviated MRI protocol was retrospectively derived from a full MRI protocol rather than prospectively optimized. In real-world practice, an a priori-designed abbreviated protocol may involve different trade-offs in acquisition parameters, such as spatial resolution, signal-to-noise ratio, and acquisition time, which affect diagnostic performance.

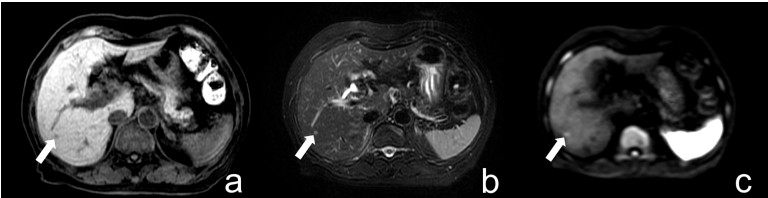

**Fig 3. False positive.** Abbreviated non-contrast liver MRI of an 83-year-old female with a focal liver lesion. (a) fat-suppressed T1WI, (b) fat-suppressed T2WI, and (c) high b-value DWI. A lesion in segment 6 (indicated by arrows) demonstrates low signal intensity on T1WI, high signal intensity on T2WI, and high signal on DWI, raising suspicion for CRLM. One observer classified the lesion as metastatic, whereas follow-up imaging revealed lesion stability, suggesting a benign etiology.

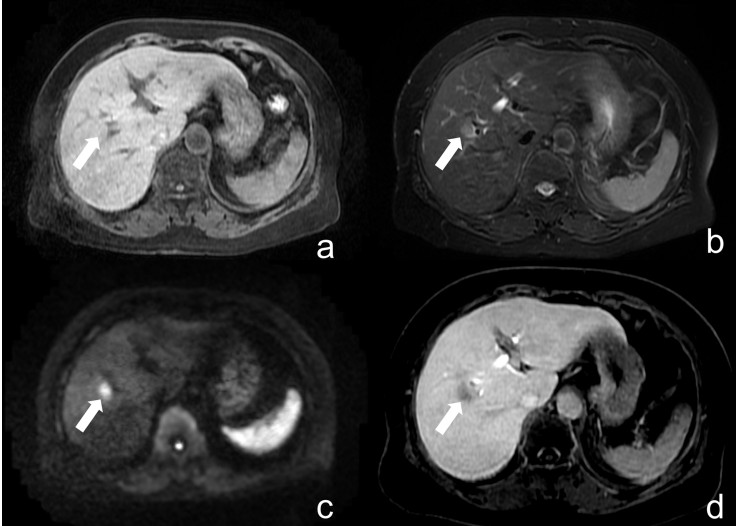

**Fig 4. False positive.** Liver MRI of a 78-year-old female with a focal liver lesion. (a) fat-suppressed T1WI, (b) fat-suppressed T2WI, and (c) high b-value DWI. A nodule in segment 8/5 (indicated by arrows) demonstrates faintly low signal intensity on T1WI, high signal intensity on T2WI, and restricted diffusion on DWI, leading both observers to classify it as a liver metastasis. **(d)** HBP image from the full contrast-enhanced MRI revealed a tract-like lesion adjacent to the nodule, exhibiting an HBP defect and mild enhancement on equilibrium phase images. The lesion was pathologically confirmed as an eosinophilic liver abscess.

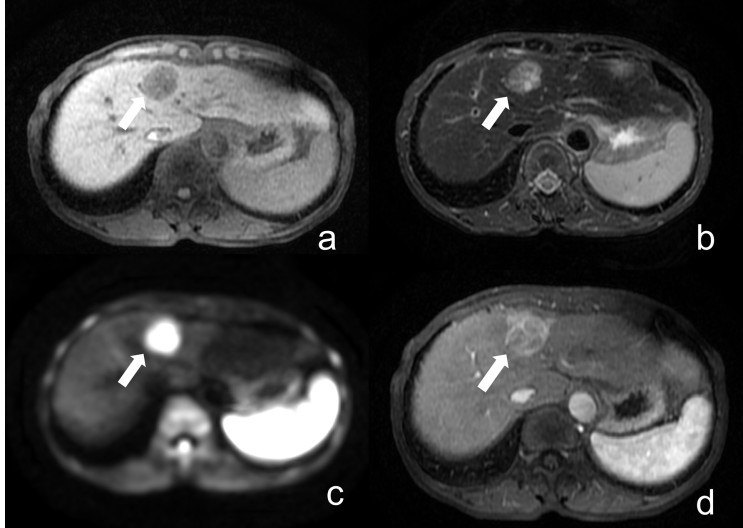

**Fig 5. False positive.** Liver MRI of a 61-year-old female with a focal liver lesion. (a) fat-suppressed T1WI, (b) fat-suppressed T2WI, and (c) high b-value DWI. A nodule in segment 8/5 (indicated by arrows) demonstrates faintly low signal intensity on T1WI, high signal intensity on T2WI, and restricted diffusion on DWI, leading both observers to classify it as a liver metastasis. (d) arterial phase T1WI shows arterial hyperenhancement of the mass, with contrast washout (not shown). The lesion was pathologically confirmed as hepatocellular carcinoma (HCC).

Fourth, examinations were performed using scanners from different vendors and with different field strengths, potentially introducing variability due to differences in hardware, acquisition techniques, and image quality. While this heterogeneity may affect internal consistency, it also reflects real-world clinical practice and may enhance the generalizability of the findings.

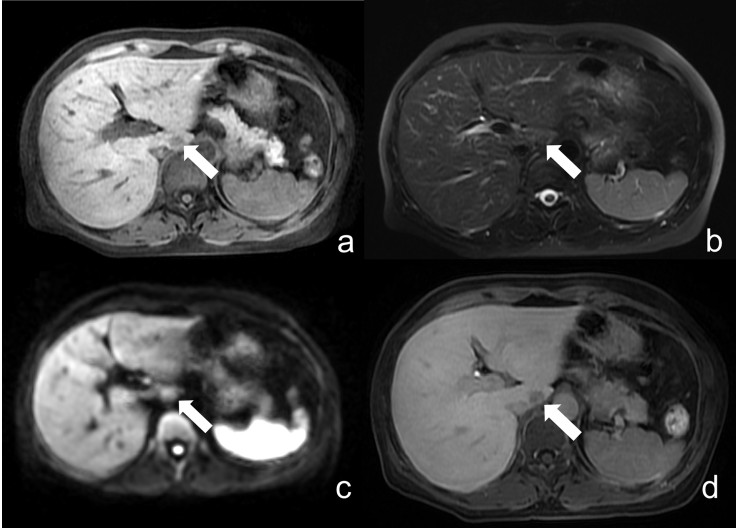

**Fig 6. False negative.** Liver MRI of a 56-year-old woman. (a) fat-suppressed T1WI, (b) fat-suppressed T2WI, and (c) high b-value DWI. A lesion in the caudate lobe (indicated by arrows) demonstrates low signal intensity on T1- and T2WI, with faintly hyperintense signal on DWI, and was initially classified by both observers as non-metastatic. **(d)** HBP image from contrast-enhanced MRI confirms the lesion as a small metastasis in the caudate lobe.

Fifth, verification bias is possible, as the reference standard mainly relied on contrast-enhanced MRI interpreted by experienced radiologists instead of on histopathology in all cases. Although histopathology remains the gold standard, radiologic assessment is widely accepted for diagnosis and treatment planning in clinical practice.

Finally, lesion size may influence diagnostic performance, as larger metastases are more easily detected, whereas smaller lesions are more likely to be missed [18]. Future studies should consider per-lesion analysis and further evaluate the impact of lesion size. Prospective and multi-institutional validation studies are warranted to confirm the role of non-contrast MRI in clinical practice.

## 5. Conclusions

This study demonstrated that abbreviated non-contrast liver MRI provides good diagnostic performance and strong interobserver agreement for detecting CRLM at primary staging. Both experienced and in-training radiologists achieved high sensitivity and specificity, supporting the clinical feasibility of this approach.

Although contrast-enhanced MRI remains the reference standard, non-contrast MRI may serve as a viable alternative in selected clinical scenarios, particularly when contrast administration is contraindicated or not feasible in resource-limited settings.

Further large-scale, prospective, and multi-institutional studies are warranted to validate these findings before broader clinical implementation. During early clinical adoption, careful image review is essential, and additional imaging should be considered in cases of uncertainty to ensure diagnostic confidence.

## Supporting information

**S1 Fig. Patient selection flowchart.** A total of 454 patients with CRC were identified through an imaging database search. Among them, 104 patients underwent liver MRI at primary staging. After excluding 17 patients (16 with other active malignancies and one with prior liver surgery or invasive liver procedures), 87 patients were included in the final analysis.
(TIF)



**S2 Fig. Receiver operating characteristic (ROC) curves for detection of CRLM.** ROC curves were generated using the 4-point diagnostic confidence scores for each reader.
(TIF)

**S1 Table. Contingency tables for the diagnostic performance of each reader.** Two-by-two tables showing classification of metastasis and non-metastasis by each reader compared with the reference standard.
(DOCX)

**S2 Table. Analysis of false-positive (FP) and false-negative (FN) cases.** Detailed description of FP and FN cases for each reader, including lesion characteristics and final diagnosis.
(DOCX)

**S1 Data. Original dataset used for analysis.** De-identified dataset provided in.dta (Stata) and.xlsx (Excel) formats.
(ZIP)

## Author contributions

**Conceptualization:** Puwitch Charoenchue, Kamon Rajchakom.

**Data curation:** Kamon Rajchakom, Wittanee Na Chiangmai.

**Formal analysis:** Puwitch Charoenchue, Kamon Rajchakom, Tanin Lertsiriladakul.

**Investigation:** Amonlaya Amantakul, Suwalee Pojchamarnwiputh, Chanon Thanaphakpaisarn.

**Methodology:** Puwitch Charoenchue, Kamon Rajchakom.

**Supervision:** Puwitch Charoenchue, Suwalee Pojchamarnwiputh.

**Visualization:** Puwitch Charoenchue, Kamon Rajchakom.

**Writing – original draft:** Puwitch Charoenchue, Kamon Rajchakom.

**Writing – review & editing:** Puwitch Charoenchue, Suwalee Pojchamarnwiputh.

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
