## [Decision Letter · Decision Letter 0]

23 Mar 2026

PONE-D-25-68814Diagnostic Performance of Abbreviated Non-contrast Liver MRI for Detecting Synchronous Colorectal Liver MetastasesPLOS One

Dear Dr. Charoenchue,

Thank you for submitting your manuscript to PLOS ONE. After careful consideration, we feel that it has merit but does not fully meet PLOS ONE’s publication criteria as it currently stands. Therefore, we invite you to submit a revised version of the manuscript that addresses the points raised during the review process.

The manuscript has been reviewed by two experts in the area. Both thought the work was both well conducted and well described, but they came to quite different conclusions. The proposed reject was based on the availability of several other and similar work in the field. Thus you will have to argue stronger as why this work adds additional value to the collected scientific knowledge. For that reason I decided on a major revision, and I think you will find the detailed comments provided by the reviewers very helpful, should you decide to revise.

We look forward to receiving your revised manuscript.

Kind regards,

Peter Lundberg

Academic Editor

PLOS One

Journal Requirements:

Reviewers' comments:

Reviewer's Responses to Questions

**Comments to the Author**

1. Is the manuscript technically sound, and do the data support the conclusions?

Reviewer #1: Yes

Reviewer #2: Yes

2. Has the statistical analysis been performed appropriately and rigorously? 

Reviewer #1: Yes

Reviewer #2: Yes

3. Have the authors made all data underlying the findings in their manuscript fully available?

Reviewer #1: No

Reviewer #2: No

4. Is the manuscript presented in an intelligible fashion and written in standard English?

Reviewer #1: Yes

Reviewer #2: Yes

5. Review Comments to the Author

Reviewer #1: General Assessment:

Well-conducted and clinically relevant study addressing the diagnostic performance of abbreviated non-contrast liver MRI for detection of colorectal metastases (CRLM). The results are strong and generally well presented. Overall, the manuscript would benefit from minor to moderate revisions focused on wording and structure as well as providing some select extra data.

Comments:

1. Statistical Analysis (Section 2.4, lines 190–193)

The reviewer understands the authors’ intention; however, the current wording is misleading and difficult to follow. A clearer explanation of the statistical approach would improve readability.

What the authors appear to have done:

A ROC analysis was performed using the 4-point diagnostic confidence score (0–3) as an ordinal test variable to discriminate between metastasis and non-metastasis. This appropriately reflects the clinically relevant binary decision in routine practice, where confidence scores of 2–3 are typically considered positive and 0–1 negative for presence of CRLM.

Hence:

The AUROC represents the overall discriminatory ability of abbreviated MRI to distinguish patients with and without metastases.

Sensitivity and specificity calculated at the predefined threshold describe test performance at this specific operating point.

Rewriting this passage to explicitly reflect this logic would greatly improve clarity and reduce the risk of misinterpretation.

2. This is along the same vein: Diagnostic performance (lines 231 - 232)

Strong results that clearly are of clinical interest. However, wording should be adjusted, as AUROC does NOT indicated "diagnostic accuracy" but measures discriminatory performance. These two are not identical. Tests with identical AUROC can differ in accuracy (think : total area under the curve is identical, but the shape can differ substantially).

The wording should be adjusted to avoid conceptual imprecision.

3. Interobserver Agreement (lines 234 - 240)

No major criticism, just a suggestion - the authors have used an ordinal scale (0 - 3). Under these circumstances, a weighted kappa (that penalizes disagreements at the levels of 0 and 3 more than on the levels 1 and 2) could be considered rather than an unweighted kappa.

4. False-Positives and False-Negative Findings

These are only mentioned in narrative form, as far as this reviewer can see. To make the paper STARD-compliant, please provide some extra information, preferably in table form , elaborating on absolute number of false positives and negatives (per reader) as well as the classification of these findings (pathology or imaging based) - e.g. hemangioma, FNH, focal fat, etc. This information, summarized in a table should be provided, not necessarily in the main text, but certainly as supplemental information.

5. Discussion on AI / Radiomics

"Future radiomics / AI may improve specificity". This is vague and generic. No AI or Radiomics approach has been implemented in this study. The authors, if they want to include this point, as AI and Radiomics are popular right now, they may at least elaborate no HOW AI and radiomics may address the specific issues described (e.g. DWI-related signal abnormalities and artefacts) or alternatively de-emphasize this point if it only stands as an isolated placative sentence.

However, without grasping for AI or Radiomics, there are other things one could (at least additionally) address to the DWI-related issues: A more immediate and practical message would be to emphasize protocol optimization and quality assurance, especially for DWI, that are crucial in a shortened protocol. With fewer sequences in an abbreviated protocol, there is less redundancy to compensate for artefacts poor image quality. Protocol optimization would be a crucial first step to address these issues without invoking AI solutions and should be emphasized more strongly.

6. DWI speculations (Line 303).

For this clinically oriented paper it is unnecessary to speculate on why certain metastatic lesion exhibit less diffusion restriction. It is sufficient to state this as an observed fact rather than speculate on the underlying mechanisms, that are more complex and numerous than only relatively reduced cellular density.

7. Line 306 - 311

Conceptually reasonable and correct. But there are possible predictors mentioned by the authors without linking them to their own data (there is no analysis on which of their FN patients ad which risk factors). So, the authors could either present and exploratory thesis ("... in our cohort, false negatives occurred in patients), or, if this is purely speculative based on external literature (entirely valid to do so), just state that this is a conceivable hypothesis based on external data in the text.

8. Terminology: "Diagnostic Accuracy" , this time in the discussion

The reviewer is entirely aware that "diagnostic accuracy" is often used as an umbrella term in radiology, encompassing both AUROC and threshold-specific metrics. However, greater conceptual clarity would be preferable. The study evaluated the discriminatory performance of an abbreviated MRI liver for CRLM and demonstrated high sensitivity, specificity and accuracy at the predefined thresholds (e.g. optimal cutoff selected by using Youden's index).

9. Prevalence and Performance Metrics (lines 265 - 266)

The sentence :

“Nevertheless, the sensitivity, specificity, accuracy, and area under the AUROC remain clinically relevant for patients, even in scenarios of lower prevalence.”

should be reworked.

The preceding paragraph correctly explains the higher prevalence of CRLM in the study cohort. Sensitivity, specificity, and AUROC are independent of prevalence, while accuracy is strongly prevalence-dependent.

With lower prevalence (as expected in real-world, non-selected cohorts), the proportion of true negatives increases, and with unchanged specificity, overall accuracy would be expected to increase rather than decrease. Given their results, the authors could argue that accuracy may be even higher in real-world cohorts. The current phrasing undersells the findings.

10. Imaging Protocol Description (lines 280 - 292 and in the Abstract).

Factually correct, but a bit verbose and unclear. It should be more clearly stated that T2 fat-suppressed sequences were used (not T2 non-fat suppressed ). This clarification is important for reproducibility and implementation by other groups.

Overall, it would likely suffice to to note that

T2 fat-suppressed sequences increase lesion conspicuity in CRLM

and

DWI further enhances lesion detectability through exploiting increased diffusion restriction in metastatic lesions.

A detailed discussion of the underlying causes of increased diffusion is unnecessary and rather increases the risk of oversimplifying complex biological causes.

Condensing and focusing this section would improve overall clarity.

11. Limitations section

The limitations section is generally strong, but should be expanded or clarified further:

a) Retrospective Design

The limitations introduced by the retrospective design are not limited to prevalence issues but also introduce other problems such as potential selection-bias and inconsistencies in protocol and timing of examinations.

b) Post Hoc Design of the Abbreviated protocol.

The abbreviated protocol is a post hoc design - it is retroactively extracted from a larger, complete protocol. This is absolutely fine and completely in keeping with study design, but it should be acknowledged that this post hoc design represents also a significant limitation.

A real world optimized abbreviated protocol, as opposed to a post-hoc one, would potentially differ in slice thickness, coverage, b-values, acquisition planes due to different optimizations for time and quality efficiency. There could be different trade offs for SNR, spatial resolution and acquisition time that would be different in a purely A Priori designed Abbreviated protocol. The necessity of likely higher needs of reliability and quality in an abbreviated protocol have been pointed out earlier in these comments.

c) Scanner and Vendor Heterogeneity

Scanners from multiple vendors were used, as well as different field strengths. Potentially there were even differences in software versions between the scanners. This introduces heterogeneity due to e.g. differences in coil-configuration, gradient performance, fat-suppression methods, EPI train lengths, b-value implementation, artefact burden, etc.

This should be acknowledged in the paper as a limitation affecting internal validity. However, at the same time, it should be pointed out that this can function simultaneously as a strength, at this heterogeneity not only increases external validity but also more closely reflects clinical reality.

Nonetheless, this should be mentioned.

11. Conclusion

a) Wording and scope, lines 327 - 333

"High diagnostic performance" could be tempered a bit - yes, it was sufficiently powered, but a sample size of n=87 is still not enormous, especially in an enriched high-prevalence cohort. Maybe "good to excellent diagnostic performance" or, "high diagnostic performance in this selected cohort" would be alternatives.

b) The statement that "non-contrast MRI may serve as a viable alternative" is correct and acceptable, but should be qualified (e.g. "for many patients" or "for selected staging scenarios"), given the observed false negatives and false positives.

c) This reviewers main problem with the discussion stems from a single sentence. The reviewer is certain that usage was unintentional, but strongly believes wording should be changed.

line 333: "particularly when contrast administration is contraindicated or impractical."

The first part : "contraindicated" is entirely correct and appropriate. Circumstances such as renal failure, prior severe reaction to contrast, pregnancy or severe allergy are all good reasons to avoid giving contrast.

However, "impractical" is not only vague but is bordering on the inappropriate. In a cancer staging context, when exactly is giving contrast "impractical"? Busy list? Limited contrast budget? Patient late in the day?

The reviewer is convinced that any of the above mentioned reasons are absolutely not what the authors intended to express. Given the stakes in an oncological context (resectability, survivability), "impractical" sounds like cutting corners rather than a medically justified choice.

Should the authors want to name a second category beyond "contraindicated", clearly clinical considerations should be considered as an alternative, such as for example "resource-limited settings were contrast is unavailable" or similar reasonings.

Otherwise a good and contructive discussion.

Reviewer #2: This is a relatively well performed and presented work on a clinically particularly relevant topic, namely the use of non-contrast MRI for the detection of liver metastases in patients with colorectal cancer.

However, I do struggle to find the novelty in this work compared to the other published studies in the same topic (e.g. ref 16, 18 and 19). It would be of great value if the authors could explain in Introduction (and Discussion) section(s) their motivation to undertake this study when there are already numerous studies in the literature on the same topic.

Other points to consider:

-Were there any patients with mucinous colorectal cancers and if yes, how could the authors differentiate metastatic depositions in the liver from cysts and/or hemangiomas?

- Abstract, line 38: “This study evaluates” should be reworded to an “Aim” and ideally it should be identical to the one in the last paragraph in the Introduction section

- Abstract, Materials and Methods: The years of performing the study should be mentioned

- Introduction, line 72: “Kidney issues”, what is the difference with the next sentence in lines 73-74?

- Introduction, lines 93-95: Hypothesis should be moved higher up and also has to be expanded to include hypothesis regarding the secondary objective. The Aims should be positioned in the last paragraph of the Introduction

- Table 1: there is no meaning having statistical comparisons (and thus p-values) in Patient Characteristics. There was no hypothesis testing here.

- In general, the authors should try to round up numbers when it is of minor importance to have decimals (i.e., mean age 65.03 makes no sense; similarly 58.6% is similar to 59%)

- Results, line 235-236: what was the reason for presenting “other reliability metrics” when the Cohen’s kappa was already evaluated?

- Fig 3: the arrow in C is misplaced.

6. PLOS authors have the option to publish the peer review history of their article (what does this mean?). If published, this will include your full peer review and any attached files.

Reviewer #1: No

Reviewer #2: No

---

## [Author Response · Author response to Decision Letter 1]

25 Mar 2026

Dear Editor and Reviewers,

We sincerely thank you for your careful evaluation of our manuscript and for the constructive and insightful comments. We greatly appreciate the time and effort invested in improving our work. We have thoroughly revised the manuscript in accordance with all comments and believe it has been substantially strengthened in clarity, methodological transparency, and clinical relevance.

Below, we provide a point-by-point response to each comment.

Reviewer #1

We are grateful for the reviewer’s positive assessment of our study and for the detailed suggestions.

1. Statistical Analysis (Section 2.4)

Response: We thank the reviewer for this important comment. We have revised Section 2.4 to improve clarity and explicitly describe the analytical approach. We now clearly state that:

• The 4-point diagnostic confidence score (0–3) was used as an ordinal variable for ROC analysis

• AUROC reflects overall discriminatory performance

• Sensitivity and specificity were calculated at a predefined clinically relevant threshold (scores 2–3 vs. 0–1)

This revision clarifies the relationship between ordinal scoring and binary clinical decision-making.

2. AUROC vs diagnostic accuracy

Response: We agree with the reviewer and have revised the manuscript to ensure conceptual clarity. The term “diagnostic accuracy” is now used only for threshold-based metrics, while AUROC is consistently described as a measure of discriminatory performance.

3. Interobserver Agreement

Response: We appreciate this suggestion. We have revised the analysis to use quadratic weighted kappa, which better reflects the ordinal nature of the 4-point confidence scale. This change has been incorporated into the Methods and Results sections.

4. False-positive and false-negative findings

Response: We have added a detailed summary of false-positive and false-negative cases in Supplementary Table S2, including the number of cases per reader and classification based on imaging and final diagnosis. Relevant examples are illustrated in Figures 3–6. This improves transparency and aligns with STARD recommendations.

5. AI / Radiomics discussion

Response: We agree that the previous statement was overly generic. We have revised the Discussion to de-emphasize AI/radiomics and instead highlight protocol optimization and quality assurance, particularly for DWI, as more immediate and clinically applicable strategies for improving diagnostic performance in abbreviated MRI protocols.

6. DWI speculation

Response: We have removed speculative explanations regarding the biological mechanisms of diffusion restriction and now present these findings as observed imaging characteristics.

7. Predictors of false-negative cases

Response: We have revised the text to clarify that these statements are hypothesis-generating and based on prior literature, rather than derived from our dataset.

8. Terminology in Discussion

Response: We have revised the Discussion to clearly distinguish between discriminatory performance (AUROC) and threshold-based diagnostic metrics, improving conceptual precision.

9. Prevalence and performance metrics

Response: We appreciate this important clarification. The relevant paragraph has been revised to state that sensitivity, specificity, and AUROC are independent of disease prevalence, while accuracy is prevalence-dependent.

10. Imaging protocol description

Response: We have revised this section to clearly specify the use of fat-suppressed T2-weighted sequences and to simplify the description, focusing on clinically relevant aspects of lesion detection.

11. Limitations

Response: We have expanded the limitations section to include:

• Broader implications of the retrospective design (including selection bias and variability in imaging conditions)

• The post hoc derivation of the abbreviated protocol

• Scanner and vendor heterogeneity, acknowledging both its potential impact on internal validity and its contribution to real-world generalizability

12. Conclusion

Response: We have revised the conclusion to:

• Use more balanced wording (“good to excellent diagnostic performance”)

• Specify applicability in selected clinical scenarios

• Replace the term “impractical” with “not feasible in resource-limited settings.”

Reviewer #2

We thank the reviewer for highlighting important aspects regarding novelty and clarity.

1. Novelty of the study

Response: We have revised both the Introduction and Discussion to better emphasize the novelty of our work. In contrast to prior studies that primarily employed lesion-based analyses or protocol comparisons, our study uses a patient-based approach in a real-world staging context, reflecting the clinically relevant decision of determining whether liver metastasis is present.

2. Mucinous colorectal cancer

Response: We have added a statement in the Discussion noting that mucinous tumors may present diagnostic challenges due to overlapping imaging features with cystic lesions. However, no dedicated subgroup analysis was performed, and this remains an area for future investigation.

3. Abstract wording

Response: The abstract has been revised to clearly state the study aim and align with the Introduction.

4. Study period

Response: The study period (January 2014 to March 2024) has been added to the abstract.

5. Terminology (“kidney issues”)

Response: We have replaced this wording with “renal impairment” for clarity and precision.

6. Hypothesis and aims

Response: We have repositioned and expanded the hypothesis to include both primary and secondary objectives, and placed the aims in the final paragraph of the Introduction.

7. Table 1 (p-values)

Response: We agree and have removed p-values from Table 1. Patient characteristics are now presented descriptively.

8. Rounding of numbers

Response: We have revised numerical values to improve readability and avoid unnecessary precision.

9. Additional reliability metrics

Response: We clarified that weighted kappa was the primary metric, and additional agreement measures were included as supportive information.

10. Figure correction

Response: The arrow placement in Figure 3 has been corrected. I’m so appreciative of your help in correcting this radiologist-matter detail.

We sincerely thank the Editor and Reviewers again for their valuable feedback. We believe the revisions have substantially improved the manuscript and we hope it is now suitable for publication.

Sincerely,

Puwitch Charoenchue, M.D., Ph.D.

24 March 2026

---

## [Editor Report · Decision Letter 1]

24 Apr 2026

Diagnostic Performance of Abbreviated Non-contrast Liver MRI for Detecting Synchronous Colorectal Liver Metastases

PONE-D-25-68814R1

Dear Dr. Charoenchue,

We’re pleased to inform you that your manuscript has been judged scientifically suitable for publication and will be formally accepted for publication once it meets all outstanding technical requirements.

Kind regards,

Peter Lundberg

Academic Editor

PLOS One
---

## [Editor Report · Acceptance letter]

PONE-D-25-68814R1

PLOS One

Dear Dr. Charoenchue,

I'm pleased to inform you that your manuscript has been deemed suitable for publication in PLOS One. Congratulations! Your manuscript is now being handed over to our production team.

Kind regards,

on behalf of

Professor Peter Lundberg

Academic Editor

PLOS One